# Timing of emergence of modern rates of sea-level rise by 1863

Jennifer S. Walker [1,2✉], Robert E. Kopp [1,2], Christopher M. Little [3] & Benjamin P. Horton [4,5]

Sea-level rise is a significant indicator of broader climate changes, and the time of emergence concept can be used to identify when modern rates of sea-level rise emerged above background variability. Yet a range of estimates of the timing persists both globally and regionally. Here, we use a global database of proxy sea-level records of the Common Era (0–2000 CE) and show that globally, it is very likely that rates of sea-level rise emerged above pre-industrial rates by 1863 CE ($P = 0.9$; range of 1825 [$P = 0.66$] to 1873 CE [$P = 0.95$]), which is similar in timing to evidence for early ocean warming and glacier melt. The time of emergence in the North Atlantic reveals a distinct spatial pattern, appearing earliest in the mid-Atlantic region (1872–1894 CE) and later in Canada and Europe (1930–1964 CE). Regional and local sea-level changes occurring over different time periods drive the spatial pattern in emergence, suggesting regional processes underlie centennial-timescale sea-level variability over the Common Era.

[1] Department of Earth and Planetary Sciences, Rutgers University, Piscataway, NJ 08854, USA. [2] Rutgers Institute of Earth, Ocean and Atmospheric Sciences, Rutgers University, New Brunswick, NJ 08901, USA. [3] Atmospheric and Environmental Research, Inc, Lexington, MA 02421, USA. [4] Earth Observatory of Singapore, Nanyang Technological University, 639798 Singapore, Singapore. [5] Asian School of the Environment, Nanyang Technological University, 639798 Singapore, Singapore. ✉email: walker@marine.rutgers.edu

The time of emergence (ToE) identifies when a climate change signal emerges above background variability, reflecting the onset of significant periods of change[1–3]. ToE has been evaluated for measures such as surface air temperatures, atmospheric variables, sea surface temperatures (SST), ocean biogeochemistry, and sea level[1,2,4]. However, these studies are often restricted to comparisons of 21st-century projections to a background reference period of observations over decades to the last century or simulated pre-industrial climate rather than using available paleoclimate data[5]. With regard to sea level, high-resolution proxy reconstructions provide a record of pre-industrial variability extending through the Common Era (0–2000 CE)[6,7], which allows for a more detailed analysis of ToE.

The goal of identifying the timing of modern rates of sea-level rise has been pursued using several different techniques and datasets. Although there is agreement that rates of sea-level rise globally and in many locations exceed Common Era background rates by the late 19th to early 20th century[8–10], global and regional estimates differ and the spatial variability in the timing among locations is unclear. Using a global tide-gauge compilation beginning in 1870 CE, Church and White[8,11] found that rates of global-mean sea-level rise were already accelerating in the late 19th century. A global sea-level reconstruction using Monte Carlo Singular Spectrum Analysis of tide gauge records suggests an acceleration starting at the end of the 18th century[12], but only a limited number of tide gauge records, restricted to northwestern Europe, extend back through to the 18th and early 19th centuries. Kopp et al.[9] estimated global sea-level change through the Common Era by applying a spatiotemporal hierarchical model to a global database of relative sea-level (RSL) reconstructions and found a significant global sea-level acceleration that began in the 19th century. From the combination of tide gauge observations and proxy reconstructions, the Sixth Assessment Report (AR6) of the Intergovernmental Panel on Climate Change (IPCC) concluded that a sustained increase of global mean sea level began between 1820 and 1860[3].

At a regional scale, Gehrels and Woodworth[10] used tide gauge and proxy reconstructions from the North Atlantic and Australasia to suggest an increase in the rate of rise between 1895 and 1945 CE; however, this estimate was simply made by visual inspection of sea-level trends. Previous studies on the U.S. Atlantic coast have used change point analysis to examine the timing of increased rates of rise at individual locations, quantifying common timing among western North Atlantic proxy records to 1865–1873 CE[13]. At individual sites, change point analysis and linear regression have identified increases in rate varying in timing from the early 19th to the early 20th centuries[7,14–17]. However, the change-point model is limited in that it assumes acceleration is instantaneous.

The range of suggested timings both globally and regionally may reflect the method of analysis or the influence of the long-term rate of change and the amplitude of pre-industrial variability among locations or may arise from regional variability in sea-level change, due to processes such as the gravitational, rotational, and deformational fingerprints of mass loss from ice sheet and glacier melt or changes in ocean and atmosphere circulation[3,18]. Concurrent analysis of sea-level change, both globally and at many individual sites, is needed to identify the source of discrepancies among estimates of the timing of increased rates of sea-level rise and to determine when sea-level rates emerged to be clearly distinguishable from background variability.

Here, we use a previously published global database (Supplementary Fig. 1, Source Data) of instrumental and proxy (e.g., foraminifera, diatoms, testate amoebae, coral microatolls, archeological evidence, sediment geochemistry) sea-level records of the Common Era[19] to examine the ToE of modern rates of sea-level rise above pre-industrial background variability. These records are incorporated into a spatiotemporal empirical hierarchical model[9,20] to examine magnitudes and rates of past RSL and global sea-level change with associated uncertainty, using shorter time periods (60-year rates) than the previous analysis[9,19] while still minimizing the effects of interdecadal fluctuations (Supplementary Table 4). We analyze the timing of the onset of modern rates of sea-level rise using the entire reconstructed spatiotemporal sea-level field from the global database. The ToE of modern rates of RSL rise is defined as when it is very likely ($P \geq 0.90$) that the rate of RSL change during a 60-year industrial-era period (from 1700 to 2000 CE) and all subsequent periods is greater than that of a random pre-industrial period (from 0 to 1700 CE). We improve upon previous analyses by assessing both global ToE, using the common global sea-level signal from proxy records, and local ToE at individual sites probabilistically and simultaneously, as opposed to separate studies of individual sites or an analysis of only the global signal. In addition, we evaluate the spatial variability of the ToE among individual locations in the North Atlantic, where the highest density of reconstructions is located, and we only analyze those sites with the highest resolution reconstructions. By decomposing RSL change into different spatiotemporal scales, we can identify regional anomalies in RSL, suggesting potential underlying processes contributing to the variability in ToE.

## Results

**Global ToE**. The rate of global sea-level rise emerged above pre-industrial rates by 1863 CE ($P = 0.9$; range of 1825 [$P = 0.66$] to 1873 CE [$P = 0.95$]), when the global rate of rise was $0.4 \pm 0.2$ mm/yr from 1840 to 1900 CE (Fig. 1). Over the pre-

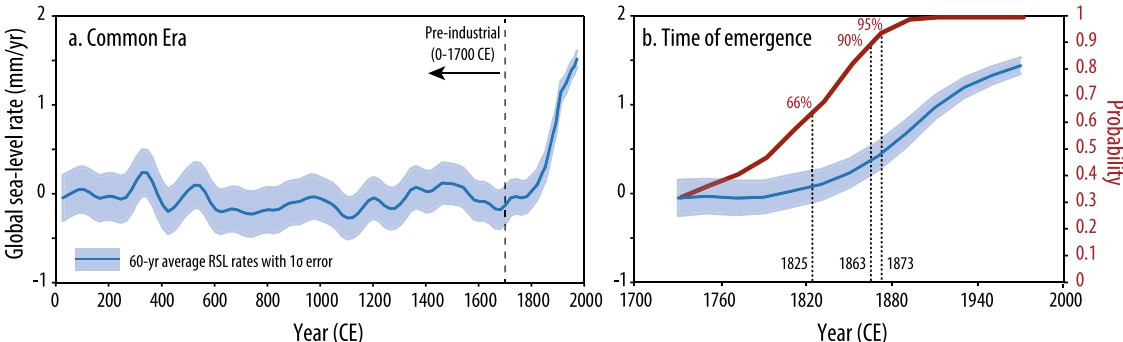

**Fig. 1 Time of emergence of global sea-level rise. a** Sixty-year average rates over the Common Era, where pre-industrial is 0–1700 CE. **b** Sixty-year average rates from 1700 to 2000 CE which increase concurrently with the probability that each 60-year interval and all subsequent 60-year intervals were greater than a random 60-year interval during the pre-industrial Common Era. The time of emergence year is given for 0.66, 0.90, and 0.95 probabilities. Model predictions are the mean with $1\sigma$ uncertainty.

**Table 1 Time of emergence and Common Era rates for global sea level, null site, and 21 North Atlantic sites.**

| Site | Time of emergence | 0–1700 CE rate (mm/yr) | 1700–2000 CE rate (mm/yr) | 1940–2000 CE rate (mm/yr) |
|---|---|---|---|---|
| Global sea level | 1863 (1825–1873) | −0.08 ± 0.05 | 0.50 ± 0.11 | 1.43 ± 0.17 |
| New York, Pelham Bay | 1872 (1840–1906) | 1.14 ± 0.09 | 2.01 ± 0.22 | 2.71 ± 0.77 |
| North Carolina, Cedar Island | 1874 (1837–1894) | 0.80 ± 0.09 | 1.79 ± 0.22 | 2.82 ± 0.78 |
| New Jersey, Cape May Courthouse | 1880 (1837–1898) | 1.44 ± 0.14 | 2.49 ± 0.24 | 3.75 ± 0.79 |
| New Jersey, Cheesequake | 1889 (1846–1906) | 1.28 ± 0.19 | 2.15 ± 0.31 | 3.21 ± 0.82 |
| North Carolina, Roanoke Island | 1894 (1842–1910) | 1.07 ± 0.05 | 1.93 ± 0.23 | 2.93 ± 0.82 |
| Florida, Little Manatee River | 1897 (1838–1912) | 0.35 ± 0.05 | 1.34 ± 0.24 | 2.41 ± 0.85 |
| Connecticut, East River Marsh | 1897 (1855–1911) | 0.95 ± 0.06 | 1.73 ± 0.27 | 2.80 ± 0.82 |
| Massachusetts, Wood Island | 1906 (1856–1927) | 0.54 ± 0.09 | 1.27 ± 0.26 | 2.11 ± 0.82 |
| Massachusetts, Barnstable | 1908 (1859–1926) | 1.33 ± 0.25 | 2.04 ± 0.38 | 3.01 ± 0.87 |
| Connecticut, Barn Island | 1908 (1860–1926) | 0.98 ± 0.19 | 1.74 ± 0.21 | 2.80 ± 0.75 |
| Florida, Nassau | 1919 (1864–1938) | 0.40 ± 0.05 | 1.04 ± 0.26 | 2.14 ± 0.84 |
| Nova Scotia, Chezzetcook Inlet | 1930 (1895–1944) | 1.77 ± 0.12 | 2.22 ± 0.20 | 3.27 ± 0.77 |
| Null site | 1933 (1877–1961) | −0.08 ± 2.17 | 0.50 ± 2.19 | 1.42 ± 2.43 |
| Newfoundland, Big River Marsh | 1942 (1900–1959) | 0.98 ± 0.05 | 1.53 ± 0.24 | 2.39 ± 0.91 |
| Iceland, Vioarholmi | 1947 (1882–NYE) | 0.63 ± 0.07 | 1.14 ± 0.30 | 1.81 ± 0.99 |
| Quebec, Saint-Simeon | 1949 (1903–NYE) | 0.80 ± 0.10 | 1.19 ± 0.28 | 1.94 ± 0.89 |
| Newfoundland, Placentia | 1951 (1908–1967) | 0.47 ± 0.06 | 0.98 ± 0.27 | 1.86 ± 0.89 |
| Denmark, Ho Bugt | 1957 (1889–NYE) | 0.49 ± 0.11 | 0.88 ± 0.33 | 1.53 ± 0.93 |
| Scotland, Loch Laxford | 1958 (1891–NYE) | 0.06 ± 0.14 | 0.41 ± 0.33 | 1.13 ± 0.91 |
| Scotland, Kyle of Tongue | 1962 (1890–NYE) | −0.23 ± 0.20 | 0.14 ± 0.35 | 0.81 ± 0.92 |
| Greenland, Aasiaat | 1963 (1902–NYE) | 0.68 ± 0.25 | 1.02 ± 0.35 | 1.86 ± 1.09 |
| Greenland, Sisimiut | 1964 (1901–NYE) | 0.27 ± 0.24 | 0.61 ± 0.34 | 1.42 ± 1.08 |

The time of emergence is the year when the probability reaches 0.90 with an uncertainty range of a lower bound when the probability reaches 0.66 and an upper bound when the probability reaches 0.95. Rates are reported with $2\sigma$ uncertainty. Null site is an indicative generic site in northeast Asia (38°N, 127°E) with no instrumental data and far from available proxy data (see Methods). *NYE* not yet emerged.

industrial Common Era, the global component of reconstructed RSL exhibits fluctuating rates between $-0.3 \pm 0.2$ and $0.2 \pm 0.3$ mm/yr from 0 to 1700 CE. Sixty-year average rates increase into the 20th century from $-0.1 \pm 0.2$ mm/yr from 1700 to 1760 CE to $1.4 \pm 0.2$ mm/yr from 1940 to 2000 CE (Fig. 1). Consistent with prior analyses[21] it is virtually certain ($P > 0.999$) that the global rate of rise from the most recent 60-year interval, 1940–2000 CE, was faster than all previous 60-year intervals during the Common Era.

Global sea-level rise is largely driven by thermal expansion of warming ocean water and increases in ocean mass due to the melting of land-based glaciers and ice sheets[22,23]. The global ToE of modern sea-level rise (1863 [1825–1873] CE) is in concordance with the onset of warming oceans. While a global synthesis of sea surface temperatures identified a cooling trend from 1 to 1800 CE[24], instrumental records of sea surface and subsurface temperatures suggest warming from the 1870s to present[25]. Using paleoclimate records since 1500 CE, Abram et al.[5] found that sustained, significant industrial-era warming trends commenced during the 19th century, preceding the global sea level ToE. Specifically, an onset of warming of sea surface temperatures occurred in the early to mid-19th century in the Western Atlantic (1828 CE), Western Pacific (1834 CE), and Indian oceans (1827 CE). While intermediate water temperatures in some regions of the ocean interior or global ocean heat content may lag surface warming trends[26,27], the onset of increased surface temperatures may signify the initial change in ocean warming, which precedes the global sea level ToE. The global ToE of modern sea-level rise is also similar in timing to global glacier retreat. Glaciers expanded globally beginning in the 13th century and reached a maximum between the mid-16th to 19th centuries, but subsequently began to retreat in the middle to late 19th century[28–30]. It is unlikely that the Greenland and Antarctic Ice Sheets had large positive contributions to global sea-level rise prior to the 20th century[31,32]. The Greenland Ice Sheet had minimal variability in

ice-mass loss over the Common Era[32] but has experienced an acceleration in the rate of ice loss in the last century[33]. The mass balance of the Antarctic Ice Sheet is not as well constrained until the last several decades when there has been an acceleration in mass loss over time[34]. Additionally, the ToE occurs after the end of the Little Ice Age (~1850 CE), which was marked by the recovery from naturally forced climate cooling due to a series of volcanic eruptions and solar variability[35,36].

**ToE in the North Atlantic.** Local records in the North Atlantic exhibit more variability than the common global signal; therefore, the ToE of modern rates of global sea level (1863 [1825–1873] CE) is earlier than individual locations in the North Atlantic. At sites in the North Atlantic, the ToE exhibits spatial variability and ranges from the late 19th century to the mid-20th century (Table 1, Fig. 2). The spatial pattern in the ToE is unrelated to pre-industrial rate variability or proxy data resolution (Supplementary Figs. 3 and 4, Supplementary Tables 2 and 5). The ToE occurs earliest in the mid-Atlantic region (New York, New Jersey, North Carolina), where it ranges from 1872 CE (1840–1906 CE) in New York to 1894 CE (1842–1910 CE) in North Carolina. The northeastern (Connecticut, Massachusetts) and southeastern (Florida) U.S. exhibit a slightly later ToE, from 1897 CE (1838–1912 CE) on the Gulf Coast of Florida to 1919 CE (1864–1938 CE) in northern Florida. Canada (Nova Scotia, Newfoundland, Quebec) and Europe (Iceland, Denmark, Greenland, Scotland) have the latest ToE, from 1930 CE (1895–1944 CE) in Nova Scotia to 1964 CE (1901 CE–not yet emerged) in Greenland.

To highlight the underlying nonlinear regional to local signal at each site responsible for the spatial pattern of the ToE, we use the spatiotemporal model decomposition (Fig. 3) to remove the global signal (which is common to all sites) (Fig. 3b) and linear signal (which is consistent with the effects of glacial isostatic

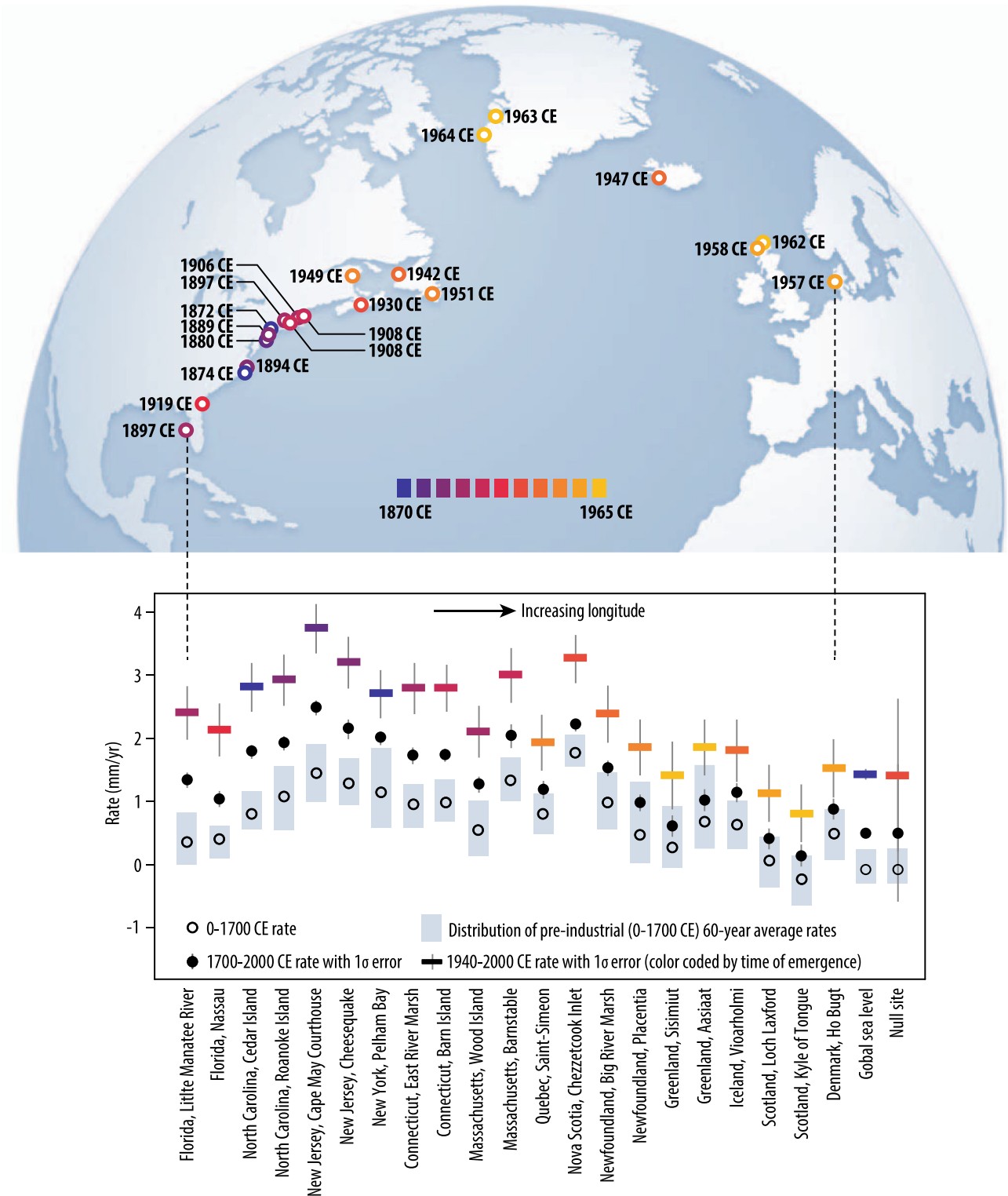

**Fig. 2 Spatial variability in the time of emergence of modern rates of relative sea level.** Time of emergence is shown for global sea level, null site (indicative generic site in northeast Asia (38°N, 127°E) with no instrumental data and far from available proxy data), and sites in the North Atlantic. Common Era rates for the periods 0–1700, 1700–2000, and 1940–2000 CE, as well as the distribution of mean estimates of pre-industrial 60-year rates, are also shown for each site by increasing longitude.

adjustment[19]) (Fig. 3c). There is a clear spatial variability of regional (global and linear signals removed) trends that correlates to the spatial pattern in the ToE (Supplementary Fig. 5). In the mid-Atlantic, it is very likely ($P > 0.9$) that the regional rates are positive for the period from 1700 to 2000 CE, with an increase beginning around 1400 CE, reaching up to $0.7 \pm 0.4$ mm/yr at the end of the 20th century in southern New Jersey (Supplementary Fig. 5a, Supplementary Table 3). The regional increase in rate is muted or absent in the northeastern and southeastern U.S. (e.g., $0.2 \pm 0.4$ mm/yr in Massachusetts and $0 \pm 0.4$ mm/yr in northern

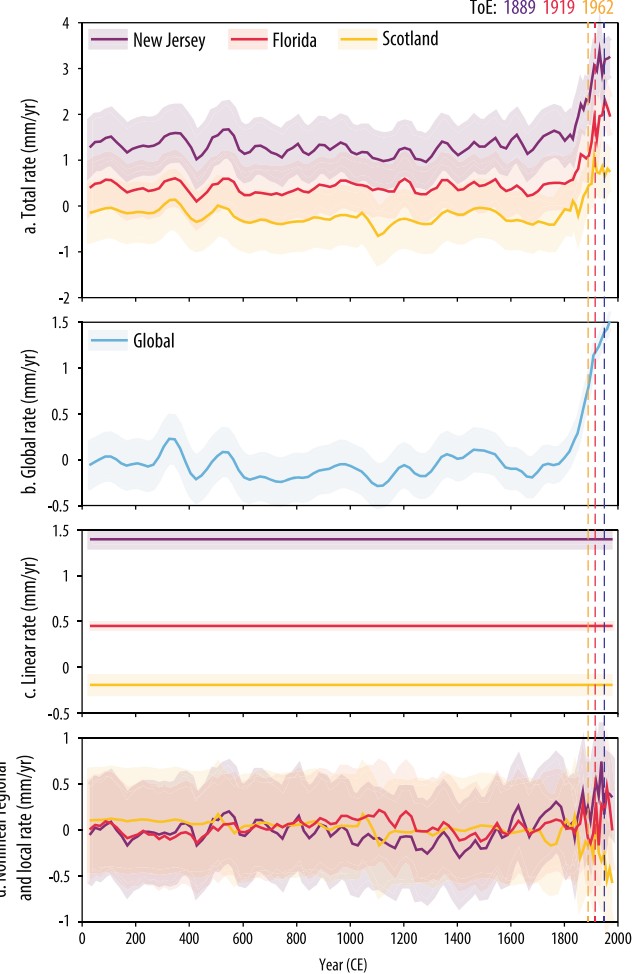

**Fig. 3 Spatiotemporal model decomposition for New Jersey, Florida, and Scotland. a** Total, **b** global, **c** linear, and **d** regional and local nonlinear rates for three sites (New Jersey, Cheesequake; Florida, Nassau; Scotland, Kyle of Tongue) are shown with the time of emergence for each site. Model predictions are the mean with 1$\sigma$ uncertainty. Note variable y-axes.

Florida at the end of the 20th century), where the probability of a positive regional rate from 1700 to 2000 CE ranges from 0.7 to 0.9 (Supplementary Fig. 5b, Supplementary Table 3). In Canada and Europe, there is no continuous pre-industrial regional rate increase to the present. In fact, it is likely ($P > 0.66$) that most sites in Canada and Europe had a negative regional rate from 1700 to 2000 CE (e.g., $-0.6 \pm 0.4$ mm/yr in Denmark and Scotland by the end of the 20th century), with a decline beginning around 1800 CE (Supplementary Fig. 5c, Supplementary Table 3).

The spatial variability in the ToE among sites suggests different centennial-timescale, regional spatial scale physical mechanisms have influenced North Atlantic sea level over the last millennium. Site-specific local mechanisms, such as increasing groundwater withdrawal causing land subsidence from growing populations at mid-Atlantic sites, could contribute to faster rates of RSL rise. For example, coastal New Jersey experienced up to ~0.7 mm yr$^{-1}$ of subsidence in the 20th century[37] and other sites on the U.S. Atlantic coast had subsidence rates in the 21st century up to double the long-term geologic rates due to groundwater withdrawal[38]. However, these are relatively recent processes and the consistency of the pattern of the ToE among and within regions suggests common regional scale driving mechanisms rather than local site-specific processes.

Along the U.S. Atlantic coast, the spatial pattern of regional rate anomalies does not suggest mass loss of the Greenland Ice Sheet, where we would expect to see an amplified RSL response to the south along the U.S. Atlantic coast[39], and therefore also a south to north pattern of the earliest to latest ToE. Previous analysis has also not detected a fingerprint of Greenland Ice Sheet melt on the U.S. Atlantic coast over the Common Era[40]. An ice sheet fingerprint may be too small or overprinted by other processes to be detectable over this time period; however, it is unlikely that the Greenland Ice Sheet significantly contributed to sea-level rise prior to the 20th century[32,33].

We thus hypothesize that the regional increase in RSL rates on the U.S. Atlantic coast is related to regional changes in the ocean and/or atmosphere. Ocean circulation changes in the North Atlantic through Atlantic Meridional Overturning Circulation (AMOC) and the strength and/or position of the Gulf Stream can affect regional sea level on the U.S. Atlantic coast[41–43]. For example, many theoretical and modeling studies have found a scaling coefficient between AMOC transport and U.S. Atlantic coast sea level on the order of $-1$ to $-2$ cm/Sv[44]. Specific climate changes coincident with the initiation of elevated regional rates of RSL rise in the mid-Atlantic have been documented in the proxy record. In particular, proxy foraminifera have been used to infer a ~3 Sv reduction in the Gulf Stream through the strength of the Florida Current during the Little Ice Age (at ~1350–1750 CE)[45]. However, sea-level rise associated with a Florida Current weakening is expected to be largest and most coherent south of Cape Hatteras;[44] therefore, the coupling of changes across the North Atlantic involving the Florida Current, Gulf Stream, and U.S. Atlantic coast sea level requires further analysis. The timing of the circulation changes and the regional increase in RSL rates in the mid-Atlantic is also roughly coincident with a shift from a sustained positive phase of the North Atlantic Oscillation (NAO) to a sustained negative phase beginning around 1400 CE[46]. Changing patterns of atmospheric winds, buoyancy fluxes, and pressure, such as those connected with the NAO[47], can affect the Gulf Stream and AMOC[48,49], which could manifest in centimeter-scale regional sea-level changes[50].

At sites in Canada and Europe, regional rates are relatively stable until ~1800 CE; however, there are several short periods of increased pre-industrial rates at several sites. Specifically, there is an increase in the regional rate between 1300 and 1500 CE, reaching 0.7–0.8 mm/yr in both sites in Greenland and one of the two sites in Newfoundland, but the rates then subsequently decrease into the 20th century (Supplementary Fig. 5). The increase in the rate in Greenland may be related to a period of growth of the Greenland Ice Sheet and crustal subsidence during this time[51]. The site in Newfoundland may experience more varied RSL trends due to its local-scale geomorphology situated on a peninsula indirectly connected to the ocean[40]. These elevated pre-industrial rates could contribute to the later ToE at these sites; however, elevated pre-industrial rates are absent at other sites within the same region that have a similar ToE, suggesting that the early elevated rates are locally driven and the ToE is substantially driven by regional processes. We thus conclude that the later ToE in Canadian and European sites is driven by a regional rate decline beginning around 1800 CE, which masks the positive rate contribution from the increasing global signal occurring during the same time. Whether the decline is related to climatic changes in the North Atlantic such as a proposed coincident AMOC weakening[52,53] is uncertain. Model evidence and theory suggest that sea-level changes associated with AMOC are smaller along the eastern boundary of the Atlantic[44,54]. There is a potential explanation outside ocean dynamics, at least for the last century: a regional sea-level fall in eastern Canada and northwestern Europe is consistent with the sea-level fingerprint of Greenland ice mass loss[55].

Overall, the ToE of modern rates of sea-level rise is later along the eastern North Atlantic margin (European coast) compared to the western North Atlantic margin (North American coast), supporting previous interpretations of differences in RSL histories between the North American and European coastlines over the last several centuries[56,57]. Therefore, while modern rates of RSL emerge at all sites by the mid-20th century due to the large magnitude common global signal ($1.4 \pm 0.2$ mm/yr by the end of the 20th century), the spatial variability in the ToE among sites is driven by asynchronous regional centennial-timescale trends unrelated to long-term linear rates of change associated with glacial isostatic adjustment (Fig. 3, Supplementary Fig. 5). Specifically, regional and local sea-level changes with an increase on the U.S. Atlantic coast since ~1400 CE and a decrease in Canada and Europe since ~1800 CE drive the spatial pattern in ToE, suggesting multiple processes underlie North Atlantic RSL variability over the Common Era. Further analysis of climate proxies and the relationship of elements of the North Atlantic circulation with RSL is needed to fully determine the cause of the regional RSL changes occurring over the last millennium, but the timing of these trends suggests that the spatial patterns in ToE are unrelated to anthropogenic climate forcing.

## Methods

**Sea-level database**. The global sea-level database[19] (Source Data) comprises Common Era RSL proxy records with high-resolution chronologies from 36 regions around the world (Supplementary Fig. 1) and includes 2274 sea-level data points from proxies such as foraminifera, diatoms, testate amoebae, coral microatolls, archeological evidence, and sediment geochemistry. In addition, decadal-average values from instrumental tide gauge records in the Permanent Service for Mean Sea Level (PSMSL;[58] were included, provided they were either (1) longer than 150 years, (2) within 5° distance of a proxy site and longer than 70 years, or (3) the nearest tide gauge to a proxy site that is longer than 20 years[9,40]. Also included are multicentury records from Amsterdam (1700–1925 CE)[59], Kronstadt (1773–1993 CE)[60], and Stockholm (1774–2000 CE)[61], as compiled by PSMSL. As in Kopp et al.[9] and Kemp et al.[40], the input data also include 1880–2010 global mean sea-level reconstruction of Hay et al.[62] from tide-gauge records.

**Spatiotemporal model**. The data is used with a spatiotemporal empirical hierarchical model[9,20]. A process-level characterizes RSL over space and time and a data level links RSL observations (reconstructions) to the RSL process.

At the process level, the RSL field $f(\boldsymbol{x}, t)$ is modeled as the sum of seven components[40]

$$f(\boldsymbol{x}, t) = g_f(t) + g_s(t) + m(\boldsymbol{x})(t - t_0) + r_s(\boldsymbol{x}, t) + r_f(\boldsymbol{x}, t) + l_s(\boldsymbol{x}, t) + l_f(\boldsymbol{x}, t) \quad (1)$$

where $\boldsymbol{x}$ represent geographic location, $t$ represents time and $t_0$ is a reference time point (2000 CE). The components are fast and slow common global terms ($g_f(t)$ and $g_s(t)$), a regional linear term ($m(\boldsymbol{x})(t - t_0)$), fast and slow regional non-linear terms ($r_f(\boldsymbol{x}, t)$ and $r_s(\boldsymbol{x}, t)$), and fast and slow local terms ($l_f(\boldsymbol{x}, t)$ and $l_s(\boldsymbol{x}, t)$). Predictions from the ICE5G–VM2–90 Earth-ice model[63] act as prior means for the regional linear term.

The data level includes the RSL reconstructions with observations, $y_i$, where

$$y_i = f(\boldsymbol{x}_i, t_i) + y_0(\boldsymbol{x}_i) + \varepsilon_i + w(\boldsymbol{x}_i, t_i) \quad (2)$$

$$t_i = \hat{t}_i + \delta_i \quad (3)$$

where $f(\boldsymbol{x}_i, t_i)$ is the true RSL value at location $\boldsymbol{x}$ and time $t$, $\varepsilon_i$ is the vertical uncertainty, $w(\boldsymbol{x}_i, t_i)$ is supplemental white noise, and $y_0(\boldsymbol{x}_i)$ is a site-specific vertical datum correction to ensure that the RSL reconstructions are directly comparable to one another. The true age of an RSL observation ($t_i$) is the mean estimate ($\hat{t}_i$) and its error ($\delta_i$).

The hyperparameters are set through a maximum-likelihood optimization to characterize prior expectations of spatial and temporal scales, as well as amplitudes, of RSL variability (Supplementary Table 1). The non-linear terms were characterized by three spatial scales (global, regional, and local) and two temporal scales (fast and slow), which allow RSL to be decomposed into global, regional temporally linear, regional non-linear, and local components. Here, we remove a constraint on the model that was applied in Kopp et al.[9], Kemp et al.[40], and Walker et al.[19] that mean global sea level over −100 to 100 CE is equal to mean global sea level over 1600–1800 CE because, with the current, expanded database, the results of the analysis with and without the constraint are nearly the same; a constant ~ $\pm 0.1$ mm/yr ambiguity in the long-term trend is no longer apparent.

**Time of emergence**. We use the reconstructed spatiotemporal field to determine the timing at which the rates in the last three centuries of the records emerge above the spread of the previous variability over the Common Era (Fig. 1). To minimize the effects of interdecadal fluctuations[12,64] and limited reconstruction resolutions, we focus on 60-year average rates. We define the pre-industrial background rates by the distribution of 60-year averages from 0 to 1700 CE at 20-year increments (e.g., 0–60 and 20–80 CE). Industrial-era rates are defined by 60-year averages from 1700 to 2000 CE at 20-year increments (e.g., 1700–1760 and 1720–1780 CE). The pre-industrial 60-year periods are enumerated from 1 to $m$, while industrial-era periods are enumerated from $m + 1$ to $m + n$. The ToE of modern rates of RSL rise is defined as when it is very likely ($P \geq 0.90$) that the rate of RSL change during industrial-era period $l$ and all subsequent periods is greater than that of a random pre-industrial period.

We define rates as $\boldsymbol{x} = [x_1, x_2, \ldots, x_m, \ldots, x_{m+n}]$, where $\boldsymbol{x}$ follows a multivariate normal distribution estimated by the statistical model. We define $\mathbf{U}$ as the $(m + n) \times (m + n)$ matrix with elements $u_{ij} = x_i - x_j$. Accounting for the full covariance among rates, we compare the industrial-era rates to the background distribution using a Monte Carlo approach, taking 10,000 samples of $\mathbf{U}$, enumerated $\hat{\mathbf{U}}_1, \ldots, \hat{\mathbf{U}}_{10000}$, with elements enumerated $\hat{u}_{ijk}$. For each Monte Carlo sample $k$, we randomly select one pre-industrial period $r_k \in [1, m]$ to serve as the reference period. For each industrial-era period $l$, we calculate $\hat{v}_{lk} = \min_{i \in [l, m+n]} \hat{u}_{ir_k k}$ and estimate the distribution of $v_l$ from the samples $\hat{v}_{lk}$. The ToE of modern rates of RSL rise is defined as the period $l$ when $P(v_l > 0) \geq 0.90$. In addition, representing each 60-year period by its central year, we interpolate the probability curve to identify the year in which the probability reaches 0.90. We provide an uncertainty range for the ToE with a lower bound of when the probability reaches 0.66 and an upper bound of when the probability reaches 0.95.

Both the highest density of RSL reconstructions from the Common Era database and the highest resolution reconstructions are in the North Atlantic. We, therefore, focus our ToE analysis on individual sites in the North Atlantic where the spatial resolution allows an examination of variability in timing. To increase the likelihood that any variability in the ToE is due to process and not the proxy data resolution, we also only analyze the ToE at proxy reconstructions in the North Atlantic that fit the following criteria: (1) the proxy record is at least ~1000 years in length to provide sufficient background information; and (2) the proxy record has data from 1700 to 2000 CE. Using these criteria, we analyze the ToE at six sites in Europe and fifteen sites along the eastern coast of North America (Table 1). In addition, we establish a "null hypothesis" by predicting RSL and the ToE at an indicative generic site in northeast Asia (38°N, 127°E) with no instrumental data and far from available proxy data to compare with the RSL and ToE at individual locations where there is available instrumental and proxy data (Supplementary Fig. 1). At this site, predicted RSL is equal to global sea level plus additional uncertainty associated with regional variability (Supplementary Fig. 2). When the RSL predictions and ToE at individual records differ from the null hypothesis, it reflects the influence of meaningful local information at individual sites.

We also conducted a series of sensitivity analyses to evaluate our ToE method. When comparing the ToE when using 40, 60, or 80-year average rates, the global ToE only varies by 15 years between the ToE using 40-year rates and 80-year rates (Supplementary Table 4). Using 40-year rates does shift the ToE at individual sites to a later date as the effects of interdecadal fluctuations become more pronounced, but the overall spatial pattern in the ToE remains broadly the same with earlier ToE in the western North Atlantic compared to the eastern North Atlantic. Using 80-year rates shifts the ToE at individual sites to slightly earlier dates, but again, the spatial pattern is largely consistent with the ToE when using 60-year average rates. Supplementary Table 5 shows the consistency of ToE when using a 0–1400 CE background reference period compared to a 0–1700 CE reference period, where the ToE globally and at individual sites remains the same. Additionally, we compared the global ToE using different datasets, including earlier versions of the proxy sea-level database from Kopp et al.[9] and Kemp et al.[40] (Supplementary Table 6). Since there is an abundance of data in the North Atlantic and western North Atlantic compared to the rest of the globe, we also compare the global ToE using only North Atlantic sites, and then all the data excluding North Atlantic sites, and all the data excluding western North Atlantic sites. In all scenarios, the global ToE still occurs in the mid to late 19th century and varies by 29 years. Finally, we tested the influence of proxy data RSL and chronological uncertainties on ToE by reducing uncertainties at European sites to be comparable to the uncertainties of proxy data on the North American coast. In this case, the ToE at European sites remained broadly the same (Supplementary Table 7).

## Data availability

Data related to this study are provided in the Supplementary Information and Source Data file. Source data are provided with this paper.

## Code availability

Code for the spatiotemporal model results that are reported in the paper[65] are available at https://doi.org/10.5281/zenodo.6030193.

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

## Acknowledgements
J.S.W. received funds from the David and Arleen McGlade Foundation. J.S.W., R.E.K., and C.L. were also supported by the U.S. National Science Foundation (awards OCE-1804999 and OCE-1805029). B.P.H. was funded by the Ministry of Education Academic Research Fund MOE2019-T3-1-004, the National Research Foundation Singapore, and the Singapore Ministry of Education, under the Research Centres of Excellence initiative and the National Sea Level Program Funding Initiative (Award USS-IF-2020-1), administered by the National Environment Agency, Singapore and supported by the National Research Foundation, Singapore. Any opinions, findings, conclusions, or recommendations expressed in this material are those of the author(s) and do not reflect the views of the N.R.F., M.N.D., and N.E.A. J.S.W. also thanks Tim Shaw for help with initial figure concepts and Muhammad Hadi Ikhsan for figure editing. The authors acknowledge PALSEA (Palaeo-Constraints on Sea-Level Rise), a working group of the International Union for Quaternary Sciences (INQUA), and Past Global Changes (PAGES), which in turn received support from the Swiss Academy of Sciences and the Chinese Academy of Sciences. This work is a contribution to IGCP Project 725 'Forecasting Coastal Change'. This work is Earth Observatory of Singapore contribution 421.

## Author contributions
J.S.W. contributed to the design of the research approach, led the research and data analysis, and wrote the first draft of the paper. R.E.K. contributed to the design of the research approach, to the data analysis, and to the writing of the paper. C.M.L. contributed to the data analysis and to the writing of the paper. B.P.H. contributed to the data analysis and to the writing of the paper.

## Competing interests
The authors declare no competing interests.
