## [Peer Review File · Nature Communications]

Timing of emergence of modern rates of sea-level rise by 1863Editorial Note: This manuscript has been previously reviewed at another journal that is not operating a transparent peer review scheme. This document only contains reviewer comments and rebuttal letters for versions considered at *Nature Communications*. Mentions of the other journal have been redacted.

REVIEWERS' COMMENTS

Reviewer #2 (Remarks to the Author):

I appreciate the authors' great efforts addressing my previous comments. I'm happy to see the publication of this study, except one last comment here,

it's good the authors have added discussions in Methods and a supplementary table about the sensitivity of ToE to the length of average rates, but it seems this is not mentioned in the main text at all. It should be briefly mentioned somewhere, maybe L92 or L111, about this sensitivity and justification for chosen 60-yr interval, to make it easier for readers receiving this information?

Reviewer #3 (Remarks to the Author):

General Comments:

This manuscript has been significantly improved since the first version, particularly the rewording to focus on the ToE and move away from the "implied" attribution.

This manuscript defines the time when global and regional (in the North Atlantic only) rates (over 60 year periods) of sea level rise emerge above the distribution of historical rates over the preindustrial period, 0-1700 CE. The time of emergence is defined as the time after 1700 when the rate exceeds 90% of the pre-1700 rates and remains above this level to the end of the record. The manuscript also gives times for the 66% and 95% probability of emergence. The global mean rates of sea level rise are the first to emerge in 1863, with a range from 1825 to 1873. Local rates emerge later, but with the earliest emergence times on the central US east coast and later times further to the north and south and significantly later in Europe. The manuscript asserts this spatial distribution is probably a result of changes in ocean circulation, mostly because other potential explanations, such as contributions to sea level rise from the Greenland ice sheet, would result in a different distribution. It is important to know when these rates emerge from the background natural variability and knowledge of the spatial distribution could potentially imply useful information on the reasons for the emergence of sea level rise. These are new and valuable results.

I remain of the view that the criteria for ToE used should be when the rates exceed 95% of historical rates and that a value of 66% is too weak a criteria for defining emergence. However, as the authors point, they give ToE for 90% and 95%, as well as the weaker 66% and I am willing to accept this.

Below are a number of essentially minor comments that I recommend the authors consider in a final revision of their manuscript. I recommend a revised manuscript will be acceptable for publication in *Nature Communications*.

Detailed Comments:

Lines 25-26: This is one definition of the times of emergence. As referee 2 pointed out, others have used a different definition for different purposes. Some rewording is required.

Line 119-120: "Suggest a warming trend from 1870s to present". This is still not quite correct. The evidence indicates the 1870s were cooler than present and not that there was a trend over the intervening period. Suggest delete the word "trend".

Lines 130-138: I do not understand these lines. Initially it is stated that Greenland and Antarctica contributed little to sea level rise. Then it states Greenland loss mass from the end of the little Ice Age and we do not know what happened in Antarctica until recent decades. And then the ToE occurs after the end of the Little Ice Age (when Greenland was contributing). Some clarification/more careful wording is required.

Line 135: where > when

Line 191: What about reference 33?

Line 362: I could not find the data or how to access it in the Supplementary Information.

Reviewer #4 (Remarks to the Author):

Overall I found this version of this manuscript [redacted] much clearer, and the authors have addressed much of the comments from the previous reviews, which has in part mitigated against my concerns that this work was too similar to previous work by the authors.

In response to my previous review, the authors have stated:

"For example, the site in southern Greenland was not included in the ToE analysis because it had minimal data spanning only ~500 years. We added a sentence in the introduction in Ln 106-107 to make this more clear"

However, I don't see this being the case, and I ask the reviewers to readdress this (I have checked through the paper to see if it was an issue with the line numbering, and I don't think it is).

Overall the set up of the paper is much clearer, and the authors are more transparent about the work this study builds upon. I do not have anything further to add and subject to the point above being addressed, I find the paper suitable for publication.

REVIEWERS' COMMENTS

Reviewer #2 (Remarks to the Author):

I appreciate the authors' great efforts addressing my previous comments. I'm happy to see the publication of this study, except one last comment here,

it's good the authors have added discussions in Methods and a supplementary table about the sensitivity of ToE to the length of average rates, but it seems this is not mentioned in the main text at all. It should be briefly mentioned somewhere, maybe L92 or L111, about this sensitivity and justification for chosen 60-yr interval, to make it easier for readers receiving this information?

In Ln 89-90, we added a statement for the justification for the chosen 60-yr interval and reference the Supplementary Table of the sensitivity of ToE to the length of average rates.

Reviewer #3 (Remarks to the Author):

General Comments:

This manuscript has been significantly improved since the first version, particularly the rewording to focus on the ToE and move away from the “implied” attribution.

This manuscript defines the time when global and regional (in the North Atlantic only) rates (over 60 year periods) of sea level rise emerge above the distribution of historical rates over the preindustrial period, 0-1700 CE. The time of emergence is defined as the time after 1700 when the rate exceeds 90% of the pre-1700 rates and remains above this level to the end of the record. The manuscript also gives times for the 66% and 95% probability of emergence. The global mean rates of sea level rise are the first to emerge in 1863, with a range from 1825 to 1873. Local rates emerge later, but with the earliest emergence times on the central US east coast and later times further to the north and south and significantly later in Europe. The manuscript asserts this spatial distribution is probably a result of changes in ocean circulation, mostly because other potential explanations, such as contributions to sea level rise from the Greenland ice sheet, would result in a different distribution. It is important to know when these rates emerge from the background natural variability and knowledge of the spatial distribution could potentially imply useful information on the reasons for the emergence of sea level rise. These are new and valuable results.

I remain of the view that the criteria for ToE used should be when the rates exceed 95% of historical rates and that a value of 66% is too weak a criteria for defining emergence. However, as the authors point, they give ToE for 90% and 95%, as well as the weaker 66% and I am willing to accept this.

Below are a number of essentially minor comments that I recommend the authors consider in a final revision of their manuscript. I recommend a revised manuscript will be acceptable for publication in Nature Communications.

Detailed Comments:

Lines 25-26: This is one definition of the times of emergence. As referee 2 pointed out, others have used a different definition for different purposes. Some rewording is required.

We reworded this sentence to illustrate that the time of emergence concept can be used in different ways.

Line 119-120: “Suggest a warming trend from 1870s to present”. This is still not quite correct. The evidence indicates the 1870s were cooler than present and not that there was a trend over the intervening period. Suggest delete the word “trend”.

We deleted the word ‘trend’.

Lines 130-138: I do not understand these lines. Initially it is stated that Greenland and Antarctica contributed little to sea level rise. Then it states Greenland loss mass from the end of the little Ice Age and we do not know what happened in Antarctica until recent decades. And then the ToE occurs after the end of the Little Ice Age (when Greenland was contributing). Some clarification/more careful wording is required.

We simplified this text to make clear that it is unlikely that Greenland and Antarctica had significant positive contributions to global sea-level rise prior to the 20th century, and there is only evidence for accelerating ice mass loss from Greenland within the 20th century or from Antarctica in the last several decades.

Line 135: where > when

We replaced ‘where’ with ‘when’.

Line 191: What about reference 33?

We added reference 33 here because it also supports this statement.

Line 362: I could not find the data or how to access it in the Supplementary Information.

We added a Supplementary Data file with the Common Era sea-level database.

Reviewer #4 (Remarks to the Author):

Overall I found this version of this manuscript [redacted] much clearer, and the authors have addressed much of the comments from the previous reviews, which has in part mitigated against my concerns that this work was too similar to previous work by the authors.

In response to my previous review, the authors have stated:

"For example, the site in southern Greenland was not included in the ToE analysis because it had minimal data spanning only ~500 years. We added a sentence in the introduction in Ln 106-107 to make this more clear"

However, I don't see this being the case, and I ask the reviewers to readdress this (I have checked through the paper to see if it was an issue with the line numbering, and I don't think it is).

The line to which we added in the introduction was "In addition, we evaluate the spatial variability of the ToE among individual locations in the North Atlantic, where the highest density and highest resolution reconstructions are located." We update this sentence to make clear that we analyze data in the North Atlantic where the highest density of reconstructions is located, but only analyze those individual sites with the highest resolution data.

Additionally, in the methods section, we have the specific criteria for sites to be included in the ToE analysis to increase the likelihood that any variability in the ToE is due to process and not the proxy data resolution.

Overall the set up of the paper is much clearer, and the authors are more transparent about the work this study builds upon. I do not have anything further to add and subject to the point above being addressed, I find the paper suitable for publication.